# MapReduce-Based D_ELT Framework to Address the Challenges of Geospatial Big Data

**Junghee Jo [1],\* and Kang-Woo Lee [2]**

[1]   Busan National University of Education, Busan 46241, Korea
[2]   Electronics and Telecommunications Research Institute (ETRI), Daejeon 34129, Korea; kwlee@etri.re.kr
\*   Correspondence: dreamer@bnue.ac.kr; Tel.: +82-51-500-7327

**Abstract:** The conventional extracting–transforming–loading (ETL) system is typically operated on a single machine not capable of handling huge volumes of geospatial big data. To deal with the considerable amount of big data in the ETL process, we propose D_ELT (delayed extracting–loading –transforming) by utilizing MapReduce-based parallelization. Among various kinds of big data, we concentrate on geospatial big data generated via sensors using Internet of Things (IoT) technology. In the IoT environment, update latency for sensor big data is typically short and old data are not worth further analysis, so the speed of data preparation is even more significant. We conducted several experiments measuring the overall performance of D_ELT and compared it with both traditional ETL and extracting–loading– transforming (ELT) systems, using different sizes of data and complexity levels for analysis. The experimental results show that D_ELT outperforms the other two approaches, ETL and ELT. In addition, the larger the amount of data or the higher the complexity of the analysis, the greater the parallelization effect of transform in D_ELT, leading to better performance over the traditional ETL and ELT approaches.

**Keywords:** ETL; ELT; big data; sensor data; IoT; geospatial big data; MapReduce

---

## 1. Introduction

In recent years, numerous types of sensors have been connected to the Internet of Things (IoT) and have produced huge volumes of data with high velocity. A large percentage of these sensor big data is geospatial data, describing information about physical things in relation to geographic space that can be represented in a coordinate system [1–4]. With the advance of IoT technologies, more diverse data have now become available, thereby greatly increasing the amount of geospatial big data.

Given the general properties of big data, the unique characteristics of geospatial data create an innovative challenge in data preparation [5]. Geospatial data typically include position data. These coordinate data differ from normal string or integer data, requiring the data pre-processing process to include a lot of floating-point arithmetic computations. Examples include transformation in geometry, converting coordination reference systems, and evaluating spatial relationships. Among these, the most well-known aspect of geospatial data is spatial relationship, describing the relationship of some objects in a specific location to other objects in neighboring locations. The calculation of spatial relationship is mostly included in spatial analysis and has been generally regarded as a sophisticated problem [6]. Moreover, processing temporal elements also complicates the handling of geospatial data.

To deal with the challenges in processing and analyzing geospatial big data, several systems have emerged. Systems designed for big data have existed for years (e.g., Hadoop [7] and Spark [8]); however, they are uninformed about spatial properties. This has led to a number of geospatial systems (e.g., SpatialHadoop [9] and GeoSpark [10]) being developed, mostly by injecting spatial data types or functions inside existing big data systems. Hadoop, especially, has proven to be a mature big

data platform and so several geospatial big data systems have been constructed by inserting spatial data awareness into Hadoop. However, it is still not easy for big data software developers to create geospatial applications. Typically, to generate a MapReduce job for a required operation in Hadoop, developers need to program a map and reduce functions. Spatial analysis usually requires handling more than one MapReduce step, where the output of the data from a previous MapReduce step becomes the input to the next MapReduce step. As the complexity level of spatial analysis is increased, the number of MapReduce steps is also increased, resulting in augmented difficulties for the developers to write iterative code to define the increasingly more complicated MapReduce steps.

To resolve this issue, in our previous work [11], we found a way to represent spatial analysis as a sequence of one or more units of spatial or non-spatial operators. This allows developers of geospatial big data applications to create spatial applications by simply combining built-in spatial or non-spatial operators, without having any detailed knowledge of MapReduce. Once the sequence of operators has been incorporated, it is automatically transformed to the map and reduces jobs in our Hadoop-based geospatial big data system. During this conversion process, our system controls the number of MapReduce steps in such a way as to achieve better performance by decreasing the overhead of mapping and reducing. The challenges for geospatial big data, however, lie in confronting not only how to store and analyze the data, but also how to transform the data while achieving good performance.

Currently, a large amount of geospatial data is continuously provided from many spatial sensors. It is important to analyze this geospatial big data as soon as possible to extract useful insights. However, the time required to transform massive amounts of geospatial data into the Hadoop platform has gradually increased. That is, it takes a lot of time to prepare the data required for geospatial analysis, thereby delaying obtaining the results of spatial analysis results. For example, we found that it took about 13 hours and 30 minutes to load 821 GB of digital tachograph (DTG) data using the traditional ETL method. In the ETL process, data are extracted from data sources, then transformed, involving normalization and cleansing, and loaded into the target data base. The conventional ETL system is typically operated on a single machine that cannot effectively handle huge volumes of big data [12]. To deal with the considerable quantity of big data in the ETL process, there have been several attempts in recent years to utilize a parallelized data processing concept [13–15].

One study [14] proposed ETLMR using a MapReduce framework to parallelize ETL processes. ETLMR is designed by integrating with Python-based MapReduce. This study conducted an experimental evaluation assessing system scalability based on different scales of jobs and data to compare with other MapReduce-based tools. Another study [15] compared Hadoop-based ETL solutions with commercial ETL solutions in terms of cost and performance. They concluded that Hadoop-based ETL solutions are better in comparison to existing commercial ETL solutions. The study in [16] implemented P-ETL (parallel-ETL), which is developed on Hadoop. Instead of the traditional three steps of extracting, transforming, and loading, P-ETL involves five steps of extracting, partitioning, transforming, reducing, and loading. This study has shown that P-ETL outperforms the classical ETL scheme. Many studies, however, have focused on big data analysis, but there have been insufficient studies attempting to increase the speed of preparing the data required for big data analysis.

In this paper, we continue our previous study on storing and managing geospatial big data and explain our approach to enhance the performance of ETL processes. Specifically, we propose a method to start geospatial big data analysis in a short time by reducing the time required for data transformation under the Hadoop platform. A transformation is defined as data processing achieved by converting source data into a consistent storage format aiming to query and analyze. Due to the complex nature of transformations, performance of the ETL processes depend mostly on how efficiently the transformations are conducted, which is the rate-limiting step in the ETL process. Our approach allows MapReduce-based parallelization of the transformation in the ETL process. Among the various sources of geospatial big data, we concentrate on sensor big data. With the increasing number of IoT sensing devices, the amount of sensor data is expected to grow significantly over

time for a wide range of fields and applications. IoT-based sensor data are, however, essentially loosely structured and typically incomplete, much of it being directly unusable. In addition, in the IoT environment, the update period—the time between the arrival of raw data and when meaningful data are made available—occurs more frequently than for typical batch data. These difficulties require that considerable resources are used for transformation in the ETL process.

This paper extends our research work presented in [11] and suggests a way to increase performance of the transformation functionality in the ETL process by taking advantage of the MapReduce framework. First, in Section 2 we briefly explain our previous work on constructing a geospatial big data processing system by extending the original Hadoop to support spatial properties. We focus particularly on explaining automatically converting a user-specified sequence of operators for spatial analysis to MapReduce steps. Section 3 describes up-to-date ETL research followed by our approach on improving performance of transformation in the ETL processes based on MapReduce. Our conducted experimental settings and results are described in Sections 4 and 5, respectively. Section 6 concludes our work and presents our plans for future research.

## 2. Geospatial Big Data Platform

In our previous study [11], we developed a high performance geospatial big data processing system based on Hadoop/MapReduce, named Marmot [17]. In Marmot, spatial analysis is defined as a sequence of RecordSetOperators, where a RecordSet is a collection of records and a RecordSetOperator is a processing element using a RecordSet, similar to a relational operator in Relational Database Management System (RDBMS). A sequence of RecordSetOperators is defined as a Plan, as shown in Figure 1.

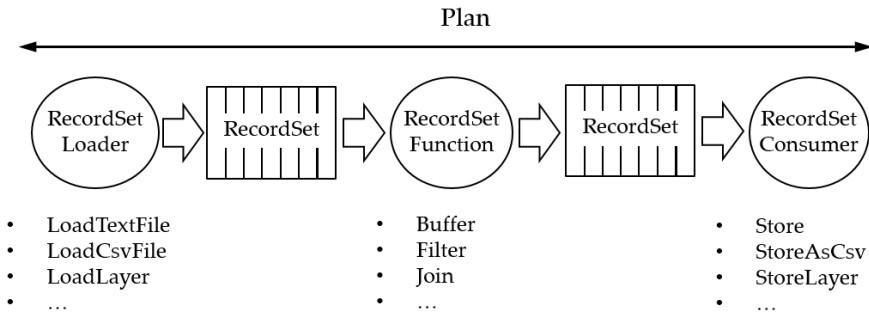

**Figure 1.** Representation of spatial analysis in Marmot: A sequence of one or more units of spatial or non-spatial operators.

In Marmot, a RecordSetOperator is classified as three possible types: RecordSetLoader, RecordSetFunction, or RecordSetConsumer. RecordSetLoader is a non-spatial operator loading source data and transforming it to a RecordSet; RecordSetFunction is a spatial or non-spatial operator taking a RecordSet as source data and producing a new RecordSet as output data; RecordSetConsumer is a non-spatial operator storing a finally created RecordSet as a result of a given spatial analysis outside of Marmot.

To process a given spatial analysis, a developer creates a corresponding Plan by combining spatial operators and non-spatial operators and injects the Plan into Marmot. Marmot processes each RecordSetOperator one by one and automatically transforms the given Plan to map and reduce jobs, as shown in Figure 2.

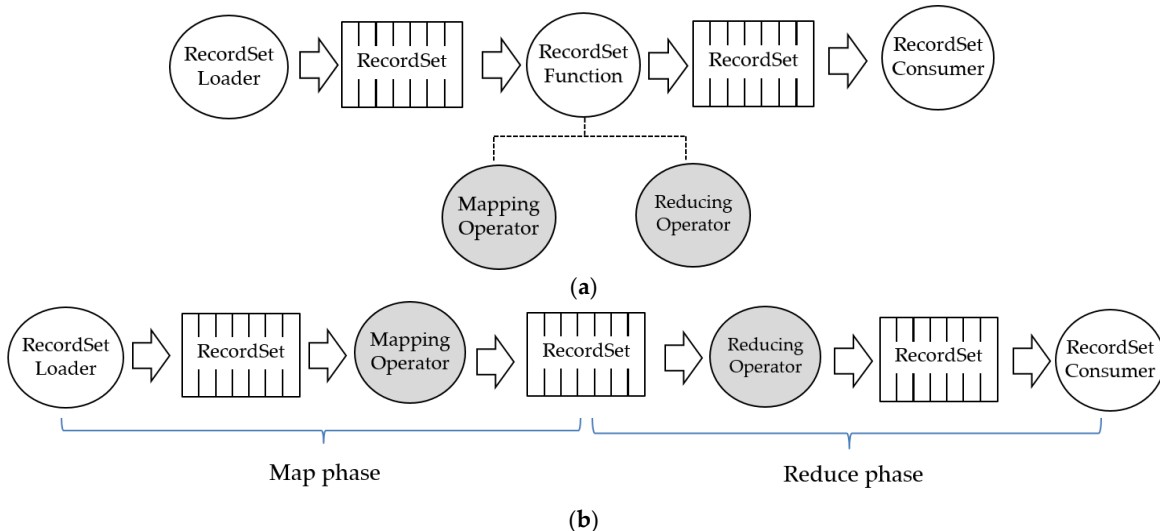

**Figure 2.** Automatic transformation of a Plan into MapReduce jobs. (**a**) A Plan having a RecordSetFunction divided into mapping and reducing operators; (**b**) An automatically transformed Plan.

While parsing a given Plan, when Marmot meets a RecordSetFunction that can be separated into mapping and reducing operators (e.g., ReduceByGroupKey), as shown in Figure 2a, Marmot decomposes the RecordSetFunction into the mapping operator and reducing operator, and eventually transforms the Plan into MapReduce jobs consisting of map and reduce phases, as shown in Figure 2b. During this transformation, Marmot controls the number of MapReduce phases in a way to achieve better performance by decreasing the overhead of mapping and reducing. To describe how Marmot handles such processes in detail, an example of spatial analysis to retrieve subway stations in a city is shown in Figures 3 and 4.

```
Plan plan;
plan = marmot.planBuilder("Subway stations per city")
            .load("logs/subway stations")
            .update("the_geom=ST_Centroid(the_geom)")
            .spatialJoin("the_geom", "region/cadastral", "the_geom",
                    INTERSECTS, "*, param.sig_cd")
            .reduceByGroupKey("sig_cd")
                .aggregate(COUNT())
            .storeAsCsv("result")
            .build();
```

**Figure 3.** An example code for searching subway stations per city.

Figure 3 is a Marmot code for an example of spatial analysis. The analysis is represented as a Plan consisting of five RecordSetOperators: Load, Update, SpatialJoin, ReduceByGroupKey, and StoreAsCsv. As shown in Figure 4, using the Load operator, Marmot reads the boundaries of each subway station and computes their center coordinates. The calculated center points are then utilized as the representative locations of each subway station via the Update operator. For each subway station, using the SpatialJoin operator, Marmot identifies the city that is the center point of the subway station. Finally, the number of subway stations per city is calculated via the ReduceByGroupKey operator and the results are stored in a CSV file named "result" via the StoreAsCsv operator.

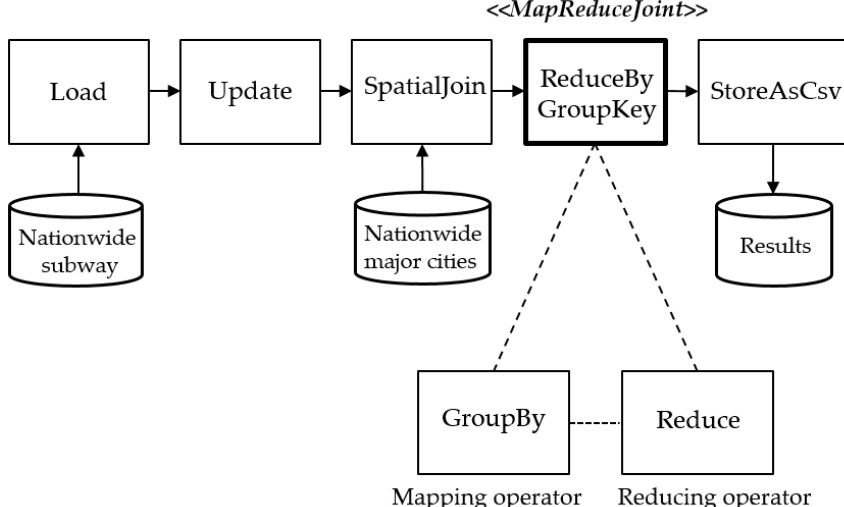

**Figure 4.** An example Plan for searching subway stations per city.

During the process of transforming the Plan to a sequence of MapReduce jobs, ReduceByGroupKey is decomposed into GroupBy and Reduce as a mapping operator and a reducing operator, respectively. Accordingly, Load, Update, SpatialJoin, and GroupBy are executed during the Map phase; Reduce and StoreAsCsv, during the Reduce phase.

## 3. Our MapReduce-Based D_ELT Framework

As mentioned in the previous section, we constructed the Marmot, high-performance data management system that enables developers with no specific knowledge of big data technologies to implement improved performance spatial analysis applications to geospatial big data. The issues concerning geospatial big data, however, lie not only in how to efficiently manage the data for fast analysis, but also in how to efficiently transform the data for fast data preparation.

DTG data, for example, have been used to analyze the status of transportation operations to identify improvement points and to identify disadvantaged areas in terms of public transportation. Transportation authorities, e.g., the Korea Transportation Safety Authority, collect DTG data from commercial vehicles and apply analytics to such big data to extract insights and facilitate decision making. Often, the results of data analysis must be derived periodically within a specific time, e.g., every single day, to be prepared for emergent cases. In this situation, to complete the given analysis in time, not only the data analysis speed, but also the data preparation speed is a critical factor affecting the overall performance. In the IoT environment, update latency for sensor big data, the focus of this paper among various sources of geospatial big data, is typically short and old data are not worth further analysis, making data preparation speed even more important. Moreover, sensor big data is machine-generated; therefore, the source data contains more noise or errors compared to human-generated data, complicating data preparation even more.

Traditional ETL [18–20] can no longer accommodate such situations. The ETL is designed for light-weight computations on small data sets, but is not capable of efficiently handling massive amounts of data. Figure 5a describes the data preparation and analysis in the ETL process. In this approach, data are extracted from various sources and then transformed on an ETL server, which is typically one machine, and loaded into a Hadoop distributed file system (HDFS). The loaded data are finally analyzed in a big data platform for decision-making. In this approach, an *analysis* operation is processed in a parallel/distributed way using MapReduce [21,22], which guarantees reasonable performance, but bottlenecks can occur during a *transform* operation. In fact, *transform* is the most time consuming phase in ETL because this operation includes filtering or aggregation of source data to fit the structure of the target database. Data cleaning should also be completed for any duplicated data, missing data,

or different data formats. Moreover, in big data environments, due to heterogeneous sources of big data, the traditional *transform* operation will create even more computational burdens. The overall performance of the ETL processes, therefore, depends mainly on how efficiently the *transform* operation is conducted.

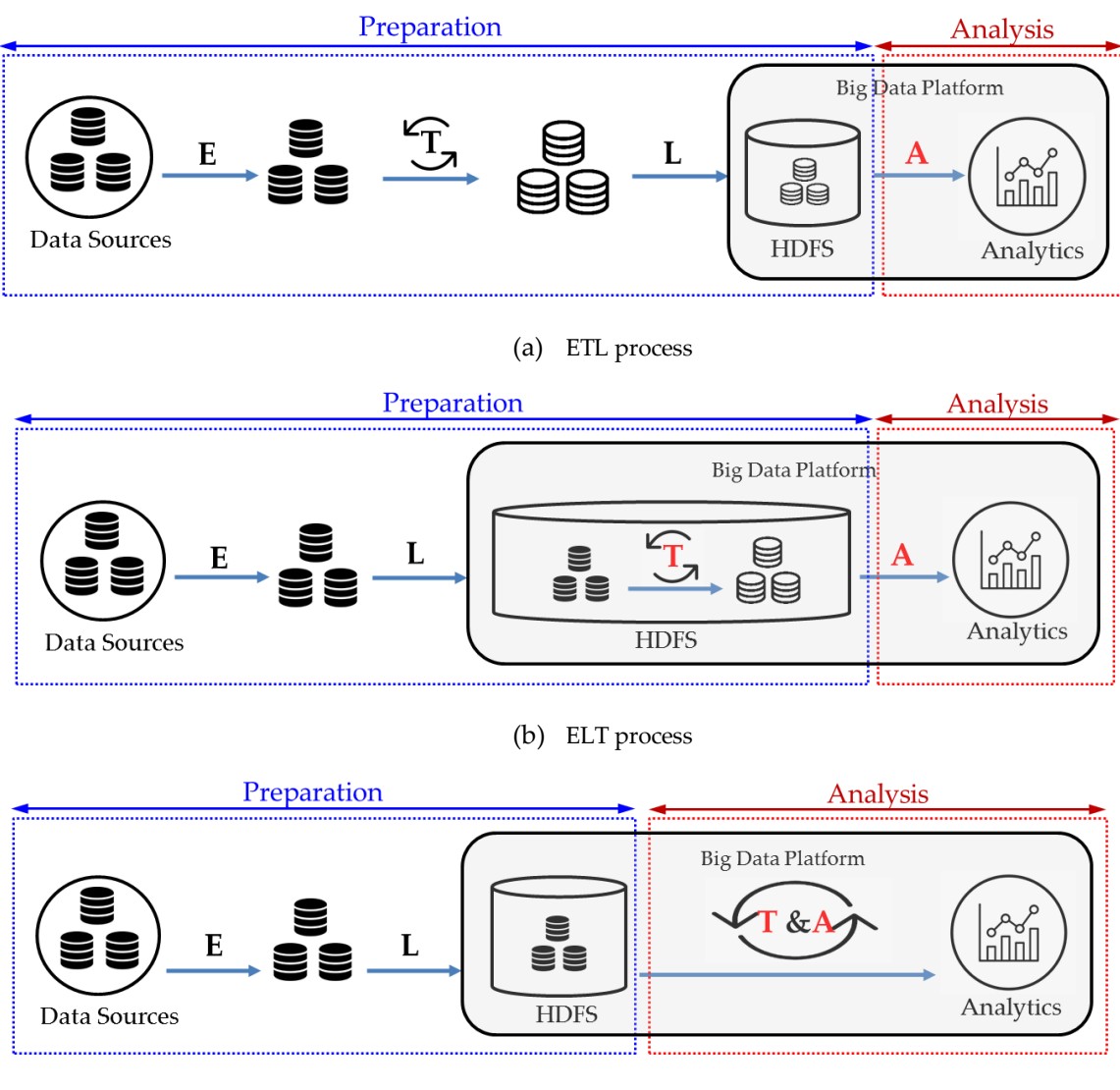

**Figure 5.** Illustration of geospatial big data preparation and analysis processes comparing three cases: (**a**) ETL; (**b**) ELT; (**c**) D_ELT. In the figure, "E" stands for extract, "T" stands for transform, "L" stands for load, and "A" stands for analysis.

To overcome the drawbacks of traditional ETL and to speed up the data preparation process, the processes of ELT was devised [23–25]. The nature of traditional ETL is to perform *transform* immediately after the *extract* operation and then start the *load* operation. In contrast, the basic idea of ELT is to conduct the *load* operation immediately after the *extract* operation, and perform the *transform* after storing the data in the HDFS, as shown in Figure 5b. This approach has several advantages over ETL. The *transform* operation can be done at the run time when needed and it is possible to use *transform* even multiple times to handle changing requirements for data. In addition, this approach eliminates a separate transformation engine, the ETL server, between the source and target and makes the overall system less costly. Above all, ELT allows raw source data to be loaded directly into the target and

also leverages the target system to perform the *transform* operation. In that sense, ELT can speed up *transform* using parallelization/distribution supported in the Hadoop-based big data platform.

Despite these advantages, ELT still has limitations in handling big data. The ELT framework can speed up *transform* using MapReduce, but *analysis* is initiated only after the *transform* has been completed. In this approach, it is difficult to optimize *transform* in conjunction with *analysis* because the *transform* is performed in a batch regardless of the context of *analysis*. For example, in the case of geospatial data, one of the high computational overheads in conducting *transform* occurs during type transformation, such as converting the x–axis and y–axis of plain-text into (x,y) coordinates of the point and coordinate system transformation for conducting spatial analysis. If *analysis* does not require such tasks, it is possible to identify them at the *transform* phase and load only the required data. By doing so, the system can eliminate unnecessary transformations and speed up performance.

To achieve better scalability and performance in conducing *transform* on geospatial big data, this paper offers a new approach for data preparation called D_ETL—in the sense that the decision of how to perform *transform* is delayed until the context of *analysis* is understood. As shown in Figure 5c, in our approach, *transform* is executed in parallel/distributed with *analysis* within our geospatial big data platform, Marmot. In Marmot, the operators for *transform* are considered a type of RecordSetOperator and are also composed of a Plan, along with the existing RecordSetOperator designed for *analysis*. This approach has the advantage that data preparation and analysis processes are described using the same data model. Application developers, therefore, can be free from the inconvenience of having to get used to implementing both processes.

Regarding the operators required to conduct *transform*, the application developer specifies them in the D_ELT script. In this way, the developer can implement both data preparation and analysis simultaneously, without having to modify the existing code for conducting *analysis*. The D_ELT script consists of the names of operators and a list of the key-values of the parameters, as shown in Figure 6. For convenience, if a developer needs a new operator for conducting *transform*, the operator can be separately implemented as a form of plug-in and can be used in Marmot, in the same way as for existing operators.

```
{
  "name": "import_plan",
  "operator": [{
    "parseCsv": {
      "delimiter": ",",
      "options": {
        "headerColumn": ["car_no", "ts", "month", "sid_cd", "besselX", "besselY",
        "status", "company", "driver_id", "xpos", "ypos"],
        "commentMarker": "#"
      }
    }
  }, {
    "expand": {
      "column": [{
        "name": "status"
      }]
    }
  }, {
    "toPoint": {
      "xColumn": "xpos",
      "yColumn": "ypos",
      "outColumn": "the_geom"
    }
  }, {
    "transformCrs": {
      "geometryColumn": "the_geom",
      "sourceSrid": "EPSG:4326",
      "targetSrid": "EPSG:5186"
    }
  }, {
    "project": {
      "columnExpr": "the_geom,*−
{the_geom,xpos,ypos,besselX,besselY,month,sid_cd}"
    }
  }]
}
```

**Figure 6.** An example of D_ELT (delayed extracting–loading –transforming) script describing operators required for data transformation.

To perform a spatial analysis, Marmot first loads the D_ELT script to determine what operators need to be executed for *transform*. Then, Marmot (1) examines the operators needed to be executed for *analysis*, (2) loads only the required data based on the need of *analysis*, and (3) executes both *transform* and *analysis* in a parallel distributed way. At this time, part of the transformed data can be used for *analysis* and not have to wait for all the data to finish being transformed. Figure 7 shows the sequence of operators executed for *transform* and *analysis* and their composition as a form of a plan. In this example Plan, "ParseCSV" is the operator for *transform* and "Filter" is the operator for *analysis*. They are allocated in the Map phase and executed in a parallel distributed way. The outputs from the Map phase are combined during the Reduce phase and the results are written in the output file.

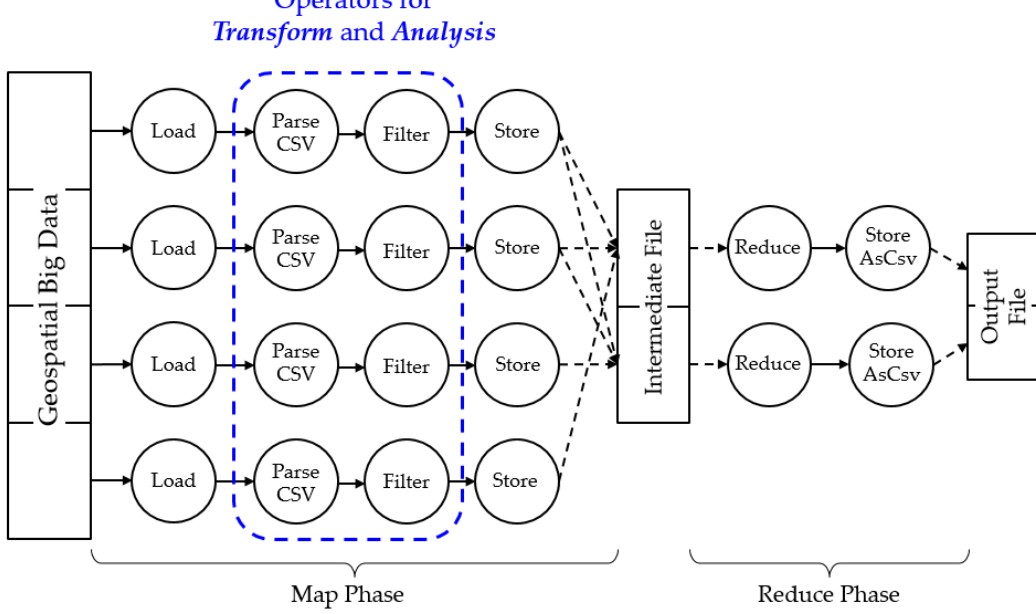

**Figure 7.** Illustration of the Map and Reduce phases during the D_ELT process.

The reason why we implemented D_ELT using MapReduce instead of Spark, another well-known engine for big data processing, is that our previously developed geospatial big platform is based on Hadoop and we had the goal of improving the data transformation time in that environment. In addition, the data we are currently handling is a large amount of DTG data, generating 20–30 TB every month. Using Spark, when running spatial analysis based on this large size of data, we anticipated that unexpected problems may occur (e.g., disk swapping), but to our knowledge, concrete solutions have not yet been proposed.

It is also important to note that ELT and D_ELT are identical in terms of performing data transformation during the MapReduce phase in Hadoop. The difference between ELT and D_ELT is as follows. In ELT, once raw data are uploaded to Hadoop, the data are transformed using MapReduce. After completely finishing the transformation, analysis is then started using another MapReduce. In D_ELT, however, data transformation is not conducted, although all of the raw data are uploaded to Hadoop but delayed until the time of conducting the analysis. That is, the transformation task is piggybacked onto the analysis task and both tasks are performed together using the same MapReduce. In this way, part of the transformed data can be used for analysis immediately without having to wait for all the data to be transformed.

## 4. Experimental Evaluation

This section explains our evaluation of the improvement in performance achieved by our proposed approach, D_ELT. In addition, the scalability of three different approaches (traditional ETL, ELT, and D_ELT) were measured and compared by varying data size and levels of analysis complexity.

### 4.1. Experimental Setup

Our experiments were conducted on the four nodes of a Hadoop cluster. Each node was a desktop computer with a 4.0 GHZ Intel 4 core i7 CPU, a 32 GB main memory, and a 4 TB disk. The operating system was CentOS 6.9 and the Hadoop version was Hortonworks HDP 2.6.1.0 with Ambari 2.5.0.3. PostgreSQL 9.5 was used for the database management system along with Oracle JDK 1.8. The 2.7.3 version of MapReduce2 was used.

The test data used in the experiment were DTG data installed in vehicles, which record the driving record in real time. The structure of the data consisted of timestamp, vehicle number, daily mileage, accumulated mileage, speed, acceleration, RPM, brake, x_position, y_position, and angle. The data were classified into three different sizes: small, 9.9GB; medium, 19.8 GB; and large, 29.8 GB, as shown in Table 1. For the geospatial big data platform, we used our developed system Marmot, as explained in Section 2.

**Table 1.** Data size: small, medium, and large.

|         | Data Size |
|---------|-----------|
| Small   | 9.9 GB    |
| Medium  | 19.8 GB   |
| Large   | 29.8 GB   |

### 4.2. Experiment 1: Measurement of Data Preparation Time

In this experiment, we compared the data preparation time of ETL and ELT to our proposed D_ELT, and the scalability of each approach based on the different data size. The overall results from this experiment are presented in Table 2.

As shown in Figure 5 in Section 3, the total time for data preparation in the ETL process includes time for extracting, transforming, and loading. In the case of ELT, the total time spent on data preparation is the summation of times for extracting, loading, and transforming. While in the ETL process, *transform* was conducted on a single machine, which is based on non-MapReduce and *transform* in the ELT was performed in a parallel distributed way based on MapReduce.

**Table 2.** Data preparation time (in seconds): ETL, ELT, and D_ELT.

|        | ETL [1] | ELT [2] | D_ELT [3] |
|--------|---------|---------|-----------|
| Small  | 579     | 413     | 116       |
| Medium | 1158    | 808     | 231       |
| Large  | 1727    | 1175    | 345       |

[1] Data preparation time in ETL: E+T+L, where E for extract, T for transform, L for load; [2] Data preparation time in ELT: E+L+T, where T is executed in a parallel distributed way; [3] Data preparation time in D_ELT: E+L.

In the case of D_ELT, the total time spent on data preparation is the summation of times only for extracting and loading, but does not include the time for transforming. *Transform* was simultaneously executed along with *analysis* in the data analysis phase, and so that the data preparation in D_ELT does not include *transform* but only *extract* and *load*.

### 4.3. Experiment 2: Measurement of Data Analysis Time

In this experiment, we compared the data analysis time of ETL, ELT, and our proposed D_ELT, and the scalability of each approach based on the different data size. Additionally, this experiment also included optimized D_ELT, D_ELT_Opt, conducting data analysis by filtering and using only the required data. In order to see the variation in performance according to the different complexity levels of analysis, we utilized three analyses—Count, GroupBy, and SpatialJoin—for low, middle, and high-level complex analysis, respectively. The overall results from this experiment are presented in Table 3.

**Table 3.** Data analysis time (in seconds): ETL(or ELT), D_ELT, and optimized D_ELT.

|  |  | ETL(or ELT) [1] | D_ELT [2] | D_ELT_Opt [3] |
|---|---|---|---|---|
| Count | Small | 68 | 96 | 57 |
|  | Medium | 126 | 179 | 104 |
|  | Large | 181 | 257 | 143 |
| GroupBy | Small | 76 | 98 | 87 |
|  | Medium | 139 | 179 | 162 |
|  | Large | 203 | 256 | 233 |
| SpatialJoin | Small | 391 | 406 | 406 |
|  | Medium | 772 | 806 | 806 |
|  | Large | 1087 | 1190 | 1148 |

[1] Data analysis time in ETL or ELT: A, where A is executed in parallel; [2] Data analysis time in D_ELT: T+A, T, A are executed in parallel; [3] Data analysis time in optimized D_ELT: T+A, where T, A are executed in parallel using only the required data.

As shown in Figure 5 in Section 3, the total time for data analysis in ETL includes only the *analysis*, which is conducted in a parallel distributed way using MapReduce. In the case of ELT, once the data preparation is completed, the *analysis* will be conducted in the same way as for ETL. In the cases of D_ELT and optimized D_ELT, the total time spent on data analysis is the time required to execute *transform* and *analysis* in a parallel distributed way using MapReduce.

## 5. Results and Discussion

### 5.1. Results

The first experiment for measuring the data preparation time for each approach reveals the following points. As shown in Figure 8, D_ELT is about 5 times faster than ETL (116 sec vs. 579 sec for small data; 231 sec vs.1158 sec for medium data; 345 sec vs. 1727 sec for large data) and about 3 times faster than ELT (116 sec vs. 413 sec for small data; 231 sec vs. 808 sec for medium data; 345 sec vs. 1175 sec for large data), regardless of the data size. The ELT approach is about 1.4 times faster than ETL. This is because of the parallel distributed processing effect using Marmot's MapReduce when performing *transform* in ELT.

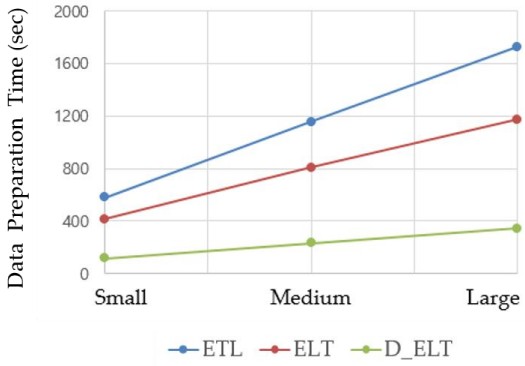

**Figure 8.** Data preparation time (in seconds): ETL, ELT, and D_ELT.

The data analysis time is measured by the second experiment and reveals the following points. Table 4 compares the performance between D_ELT and ETL(or ELT) and optimized D_ELT and ETL(or ELT). In both cases, the ratio of performance to analysis is almost identical regardless of the data size. An interesting point is that the data analysis time in the D_ELT process contains time for *transform*, while the ETL(or ELT) process does not include this time. Although D_ELT is slower than ETL(or ELT), there is little difference in performance—D_ELT is up to 1.4 times slower. In the case of optimized D_ELT, the process is only up to 1.2 times slower than the ETL(or ELT) approach. In D_ELT, in the case

of simple analysis, the time involved in data transforming is relatively large compared to the analysis time and consumes a large part of the total execution time. However, in the case of complex spatial analysis, the time involved in data transforming is relatively small compared to data analysis, and so the transformation overhead incurred is relatively small.

**Table 4.** Performance comparison: D_ELT/ETL(or ELT) and optimized D_ELT/ETL(or ELT).

|  |  | **D_ELT/ETL(or ELT)** | **D_ELT_Opt/ETL(or ELT)** |
|---|---|---|---|
| Count | Small | 1.41 | 0.84 |
|  | Medium | 1.42 | 0.83 |
|  | Large | 1.42 | 0.79 |
| GroupBy | Small | 1.29 | 1.14 |
|  | Medium | 1.29 | 1.17 |
|  | Large | 1.26 | 1.15 |
| SpatialJoin | Small | 1.04 | 1.04 |
|  | Medium | 1.04 | 1.04 |
|  | Large | 1.09 | 1.06 |

It is important to note that these have no effect on overall performance degradation, considering that D_ELT is about 3–5 times faster than ETL(or ELT) during data preparation, as shown in Table 2 in Section 4 and Figure 8. Therefore, the performance of both D_ELT and optimized D_ELT is much greater than that of ETL or ELT.

Figure 9 compares the performance between D_ELT and ETL(or ELT) during data analysis according to the different data size and analysis type. As aforementioned, the analysis time in the D_ELT includes *transform* as opposed to ETL(or ELT), and so D_ELT is slower than ETL(or ELT), as shown in Table 3 in Section 4. However, the higher the complexity of the analysis (Count < GroupBy < SpatialJoin), the smaller the difference between the D_ELT and the ETL(or ELT) performance. The reason is that the higher the complexity of analysis, the higher the effect of parallelizing the *transform* of D_ELT, thereby enhancing the performance of D_ELT.

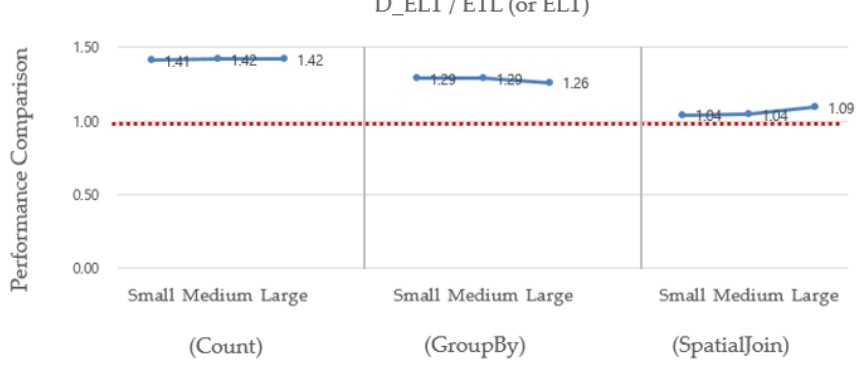

**Figure 9.** Performance comparison of D_ELT/ETL(or ELT) based on the small, medium, and large data size for each of three analyses: Count, Group-By, and SpatialJoin.

Similarly, Figure 10 compares the performance between optimized D_ELT and ETL(or ELT) during data analysis, according to different data size and analysis type. Compared to Figure 9, in the case of two simple analysis cases, Count and GroupBy, optimized D_ELT is faster than D_ELT. This is because for simple analysis, large amounts of data are often unrelated to the analysis, and so more data can be included in the optimization target, resulting in an incremental increase in performance of optimized D_ELT.

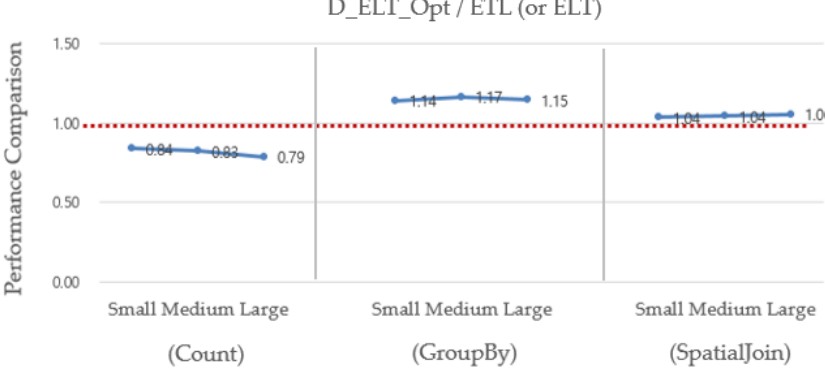

**Figure 10.** Performance comparison of optimized D_ELT/ETL(or ELT) based on the small, medium, and large data size for each of three analyses: Count, Group-By, and SpatialJoin.

In both cases, the performance ratio of the analysis is very similar regardless of data size. Thus, we chose only the small data size to compare the performance ratio, as shown in Figure 11. This shows that the higher the complexity of the analysis, the smaller the performance difference between D_ELT and optimized D_ELT. This is because the more complex the analysis, the more data is involved in the analysis, which reduces the scope of optimization. In the case of SpatialJoin, which has the highest complexity among the three analyses, the two values in Figure 11 converge to almost 1.0, showing that there is almost no performance difference between D_ELT and optimized D_ELT.

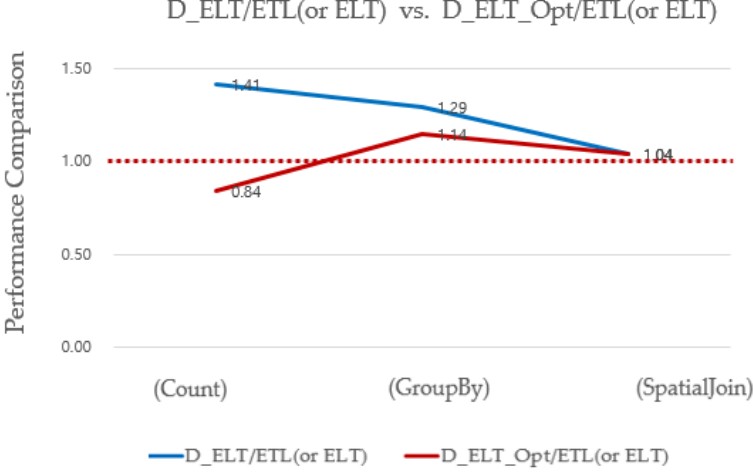

**Figure 11.** Performance comparison of D_ELT/ETL(or ELT) vs. optimized D_ELT/ETL(or ELT) based on the small data size for each of three analyses: Count, Group-By, and SpatialJoin.

The overall performance of ETL, ELT, D_ELT, and optimized D_ELT is derived by summing the data preparation and analysis times. Table 5 shows that the overall performance of D_ELT is much faster than that of the ETL or ELT approaches. D_ELT is up to 3 times faster than ETL and 2 times faster than ELT. Optimzed D_ELT is up to 4 times faster than ETL and 3 times faster than ELT. The results are derived from the two simple analysis cases, Count and GroupBy, but not SpatialJoin. In the case of SpatialJoin, both D_ELT and optimized D_ELT still perform better than ETL or ELT, but there is almost no difference between the overall performance of D_ELT and optimized D_ELT. Figure 12 shows that as the complexity of the analysis is increased, the gap between D_ELT and optimized D_ELT is decreased.

**Table 5.** Overall performance of ETL, ELT, D_ELT, optimized D_ELT (in seconds), and performance comparison among ELT vs. ETL, D_ELT vs. ETL, and optimized D_ELT vs. ETL.

|  |  | ETL (sec) | ELT (sec) | D_ELT (sec) | D_ELT_Opt (sec) | ELT/ ETL | D_ELT/ ETL | D_ELT_Opt/ ETL |
|---|---|---|---|---|---|---|---|---|
| Count | Small | 647 | 481 | 212 | 173 | 0.74 | 0.33 | 0.27 |
|  | Medium | 1284 | 934 | 410 | 335 | 0.73 | 0.32 | 0.26 |
|  | Large | 1908 | 1356 | 602 | 488 | 0.71 | 0.32 | 0.26 |
| GroupBy | Small | 655 | 489 | 214 | 203 | 0.75 | 0.33 | 0.31 |
|  | Medium | 1297 | 947 | 410 | 393 | 0.73 | 0.32 | 0.30 |
|  | Large | 1930 | 1378 | 601 | 578 | 0.71 | 0.31 | 0.30 |
| SpatialJoin | Small | 970 | 804 | 522 | 522 | 0.83 | 0.54 | 0.54 |
|  | Medium | 1930 | 1580 | 1037 | 1037 | 0.82 | 0.54 | 0.54 |
|  | Large | 2814 | 2262 | 1535 | 1493 | 0.80 | 0.55 | 0.53 |

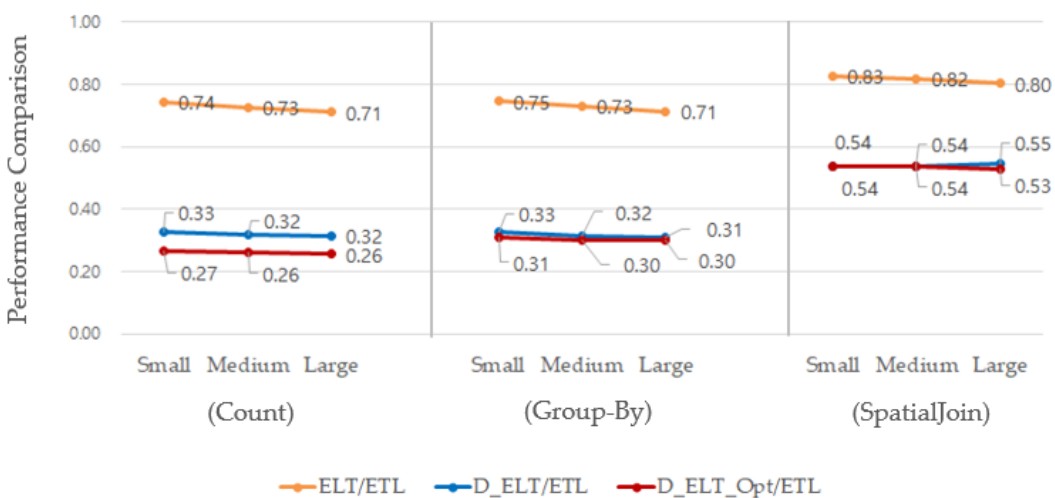

**Figure 12.** Overall performance comparison of ELT/ETL vs. D_ELT/ETL vs. optimized D_ELT/ETL based on a small, medium, and large data size for each of three analyses: Count, Group-By, and SpatialJoin.

## 5.2. Discussion

There are two conventional methods—ETL and ELT. The traditional ETL method does not use the distributed/parallel method during data pre-processing, causing problems especially when the volume of data to be pre-processed is large. The ELT method improves traditional ETL methods to speed up data pre-processing using the distributed/parallel method. Our proposed D_ELT method reduces overhead in data pre-processing. In D_ELT, the transformation task is piggybacked onto the analysis task and both tasks are performed together using the same MapReduce. This way allows one to conduct the analysis immediately without storing transform results and also excludes unnecessary transformations that are not utilized in the analysis.

Compared to existing methods, however, the D_ELT method significantly reduces the data preparation time, but has the disadvantage in the following cases. First, the case that the same kind of analysis must be conducted repetitively. For example, the D_ELT method results in a 1382-second reduction (large data, Table 2) in data preparation time compared to that of the conventional ETL method, but 103 seconds is added (large data, SpatialJoin, Table 3) every time an analysis is conducted. Therefore, the greater the number of conducting analyses, the more inefficient D_ELT is compared to traditional methods. In the example above, the D_ELT method is more inefficient than the existing method when the same analysis is conducted more than 14 times in succession. Second, in the case that a large amount of input data is invalid, a large amount of data can be removed as a result of the transform. In D_ELT, the transformation task is piggybacked every time an analysis task is executed,

a large amount of invalid data is repeatedly read, resulting in unnecessary I/O and computation burden. Finally, the method proposed in this paper does not consider real-time applications. However, it provides the advantage that required analysis results can be obtained relatively more quickly than other conventional methods.

## 6. Conclusions

This paper presents our proposed D_ELT approach to efficiently transform and analyze data, thereby making it usable for a large amount of sensor big data, especially geospatial big data. Based on the experimental results, we made several observations as follows. First, D_ELT outperforms ETL and ELT during data preparation. Second, D_ELT shows performance degradation during data analysis. However, the higher the complexity of the analysis, the smaller the performance degradation, resulting in overall improved performance compared to ETL or ELT. Finally, in the case of simple analysis increasing the scope of optimization, optimized D_ELT outperforms ELT. In the future, we plan to further increase the overall performance of our developed system including D_ELT and Marmot by investigating the spatial index, to better support spatial queries in dealing with geospatial big data.

**Author Contributions:** K.-W.L. designed and implemented the D_ELT; K.-W.L. and J.J. conducted the testing of D_ELT and analyzed the experiment results; J.J. wrote and K.-W.L. revised the manuscript.

**Funding:** This research was supported by the MOLIT (The Ministry of Land, Infrastructure and Transport), Korea, under the national spatial information research program supervised by the KAIA(Korea Agency for Infrastructure Technology Advancement) (19NSIP-B081011-06).

**Acknowledgments:** This research, 'Geospatial Big data Management, Analysis and Service Platform Technology Development', was supported by the MOLIT (The Ministry of Land, Infrastructure and Transport), Korea, under the national spatial information research program supervised by the KAIA(Korea Agency for Infrastructure Technology Advancement) (19NSIP-B081011-06).

**Conflicts of Interest:** The authors declare no conflicts of interest.

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
