# Peer review of "MapReduce-Based D_ELT Framework to Address the Challenges of Geospatial Big Data"

_ijgi, doi:10.3390/ijgi8110475_

Round 1
Reviewer 1 Report
This paper discusses a very meaningful research topic about the ETL for big data, which is the problem to be faced in the beginning of data chain. However, the algorithm and experiment are too simple to reflect innovation well. At the same time, the characteristics of Internet of things data are not well presented.
[General comments]
1、In the title, I suggest to use the ELT instead of ETL, and ELT is proposed by this paper. Is it ok?
2、In the fourth paragraph (line 56), here, in addition to introducing your team's work, I suggest adding some solutions from other international teams in this issue, which will show it is a common problem.
3、For the section 2, I don't think it would be necessary to spend a lot of words here on previous work. Moreover, some contents are duplicated, such as figure 5 with reference [11]. Compressed text paragraphs are recommended, which does not affect the integrity of the paper.
4、The big data platform/ETL algorithm and the IOT data are open resource or not?
[Detail comments]
The title of section 3, using ‘a mapreduce-based ETL framework for IOT’ or ‘a mapreduce-based ETL framework’. Regarding the amount of data, provide the data entries and the meaning of the data, or data structure, help the reader to understand the data. Table 5 is the same content to Figure 13.Author Response
Please refer to the attached file.

Reviewer 2 Report
This paper proposed a D_ELT approach to improve the big data processing performance over the existing ETL approach. The work is interesting and useful. However, I have several concerns about the work.
Detail the differences between ELT and D_ELT. I don’t see any particular innovation of D_ELT over ELT. Both are moving the T into the MapReduce phase. D_ELT uses a pre-filter to avoid unnecessary transform processes in the Map phase. But is ELT not doing the same thing? ELT is also designed to transform data for final analysis as per demand. What is the difference? Another concern is about the disadvantages of the approach. Some spatial big data is huge in size, like satellite images. The Transform process can scale them into reasonable small size to reduce the burden on Hadoop. However, since the T is moved into Hadoop, the huge images have to be entirely loaded into Hadoop which will take much more time than the reduced time by the proposed delayed transform process. Please illustrate the disadvantages of your approach like in such kind of contexts. The title says Internet of Things, but I have not seen any scenario or context about IoT background. The data and context is not well introduced. Since the D_ELT is supposed to be the original contribution, I think it should replace the ETL in the title to emphasize. Have you compared the performance with Spark? Spark is like 100x times faster than Hadoop, which you might want to try. The used spatial analysis is too simple. IoT requires a lot of other spatial processes which are more complicated, e.g., risk assessment, interpolation. I think it will be more persuasive if more spatial analysis functions are compared.Author Response
Please refer to the attached file.

Reviewer 3 Report
An interesting new approach to big geodata management, encompassing several technologies together. The paper is well written and the research carried out is cutting edge, so the paper is publishable, but the authors need to consider the following issue in proceeding to a minor revision of their paper:
Right from the abstract (line 15), the authors set complexity as the key issue to be tackled from within their research project: "To this end, to deal with the complexity 14 of big data in the ETL process, we propose D_ELT...".
Also, the word “complexity” appears as msny as 14 times in the paper and the word “complex” 3 times also.
But, contrary to what was expected, the complexity of geodata was not discussed to the length that it deserved.
Thus, the authors are advised to add a comment that the complexity of geo-data refers to both the structural/spatial context of geodata and to their functional characteristics (Papadimitiou, 2013) and it plays a central role in managing geospatial web semantics also (Perry et al, 2007) and, inevitably, the higher the geographic and/or cartographic complexity of the big geodata, the higher the times and computational resources anticipated to treat them.
Papadimitriou, F. (2013). Mathematical Modelling of Land Use and Landscape Complexity with Ultrametric Topology. Journal of Land Use Science, 8(2), 234-254.
Perry M., Sheth A.P., Hakimpour F., Jain P. (2007). Supporting Complex Thematic, Spatial and Temporal Queries over Semantic Web Data. In: Fonseca F., Rodríguez M.A., Levashkin S. (eds) GeoSpatial Semantics. GeoS 2007. Lecture Notes in Computer Science, vol 4853. pp 228-246. Springer, Berlin, Heidelberg.
Round 2
Reviewer 1 Report
According to the author's modification, it is recommended to accept!
However, on the question of title, authors are advised not to delete the Internet of things.
In addition, some references for your reference!
[1]Utilizing MapReduce to Improve Probe-Car Track Data Mining
[2]LandQv2: A MapReduce-Based System for Processing Arable Land Quality Big Data
Author Response
Please refer to the attached letter. Thank you.

Reviewer 2 Report
I am afraid my concerns are not well addressed by this revision. Most contents responding to my suggestions cann’t be found in the text. The authors need label every place changed and link the changes to their corresponding responses to the raised question.
According to the response, D_ELT is not always better than ELT, and the experiment results could be overturned in some scenarios. It will be misleading for people who face the situation that the transformation process is used to shrink the data volume. It need be explicitly explained in the discussion section about the disadvantages of the proposed approach.
The abstract says D_ELT is proposed, but the title says ELT. Which is correct?
The data and context are still totally missing. Answer questions like why you do this? What is your big data problem exactly? Why the existing data processing platforms fall short? Are the analyzed results required on near real time basis? How much data volume are you referring to in your experiments? These things are called context and will help readers to understand the value of your work. Also, the important thing that differentiates this work from other ETL research is geospatial data. What difference do the geospatial features bring to this ELT framework? Any particular changes made for spatiotemporal features in your approach? Or your approach is actually not specifically for geospatial data but can be used for any data.
In the letter, the authors said “this method can also be implemented in Spark”. That is based on zero evidence and is not convincing at all. The authors also said “However, using the proposed method in this paper, the higher the complexity of analysis, the higher the effect of parallelizing the transform, therefore, the better the result of enhancing overall performance, as described in Section 5.” Not convincing. Complex spatial analysis with high algorithm complexity will lay a lot of burden on the nodes and moving transformation process to the MapReduce phase seems not helping that.
Author Response

(The authors gave the same response as above.)

Round 3
Reviewer 2 Report
Some language issues exist in the changed text. Suggest to have native speaker read through the manuscript to correct them.
Author Response
Dear Reviewer,
We appreciate the time and thoughtful observations. For English editing, we sent our paper to a Native American in US who has official certificate for English editing service.
For your information, we would like to let you know that we have previously applied the second revisions in the paper and now only the English revisions are shown in the paper using track changes for your convenience.
Thank you, Junghee Jo